# Cross-Training and Resistance Training in Adults with Type B Acute Lymphoblastic Leukemia during the Induction Phase: A Randomized Blind Pilot Study

**DOI:** 10.3390/jcm12155008

**Published:** 2023-07-30

**Authors:** Adán Germán Gallardo-Rodríguez, Vanessa Fuchs-Tarlovsky, María Esther Ocharán-Hernández, Christian Omar Ramos-Peñafiel

**Affiliations:** 1Research in Medicine Program, Instituto Politécnico Nacional, Mexico City 11340, Mexico; nutriologo.agallardo8@gmail.com; 2Hematology Research Department, Hospital General de México “Dr. Eduardo Liceaga”, Mexico City 06720, Mexico; 3Clinical Nutrition Department, Hospital General de México “Dr. Eduardo Liceaga”, Mexico City 06720, Mexico; dravanessafuchs@gmail.com; 4Postgraduate Studies and Research Section, Escuela Superior de Medicina, Instituto Politécnico Nacional, Mexico City 11340, Mexico; estherocharan@hotmail.com; 5Hematology Department, Hospital General de México “Dr. Eduardo Liceaga”, México City 06720, Mexico

**Keywords:** exercise therapy, acute lymphoblastic leukemia, quality of life, body composition, physical performance, relapse

## Abstract

Patients with acute lymphoblastic leukemia (ALL) undergoing induction decrease their physical capacity, lose muscle mass, and decrease their quality of life (QOL). The safety, feasibility, and benefits of exercise during chemotherapy have been proven, but the effects of cross-training activities have yet to be analyzed. To measure the effects of cross-training on body composition, physical performance, and QOL, a blind randomized clinical trial was carried out. A total of 33 patients were included and randomized into a cross-training exercise group (CEG), a resistance exercise group (REG), and a control group (CG). During induction, patients received an exercise routine three to five days a week for 30 to 50 min each. Body composition, QOL, and physical performance were measured at baseline, up to discharge, and at a follow-up of two months. Body composition improved in the REG and CEG. In the CG, muscle mass decreased and fat mass increased (*p* = 0.020 and 0.020, respectively). The REG and CEG had significant positive improvements in physical performance compared to the CG. QOL showed no differences in any group (*p* = 0.340). Cross-training and resistance exercise are essential to improve body composition and physical performance during induction. Considering the prognostic value of physical performance, we propose integrated training exercises as adjuvant therapy in adult patients with ALL.

## 1. Introduction

Acute lymphoblastic leukemia (ALL) is a type of hematological cancer characterized by the differentiation, proliferation, and accumulation of damaged lymphoid progenitor cells found in the bone marrow and extramedullary areas [1]. This is the most frequent cancer found in children and young adults (1.8 cases out of 100,000 people), with an expected survival rate of 70.8% in this population [2]. In Mexico, it is considered the most frequent acute leukemia (51%), which originates from precursor B cells (93.6%) [3]. The treatment requires a combination of chemo- and immunotherapy; in specific cases, it also involves tyrosine kinase inhibitors. Induction therapy is the most important since its resistance worsens the prognosis, and this stage lasts four weeks on average.

The severity of hospital adverse events may vary, with hematological toxicity as the main event. However, other events might appear, such as skeletal muscle deterioration or neuromuscular and physical performance impairment compared to the healthy population [4]. The main structure of the treatment includes steroids, vinca alkaloids, and anthracyclines. This last one increases the risk of cardiovascular complications (arrhythmia and heart failure), neurological complications (peripheral and sensory neuropathy as well as convulsive crises), and thrombotic complications [5]. Depending on the severity, these events might delay treatment and affect the prognosis. Other secondary sources that are considered light and tolerable but might also affect the quality of life are nausea, pain, fatigue, sleep disorders, and anxiety. Consequently, quality of life is the most relevant aspect and goal for most oncology treatments [6]. Evidence indicates that exercise might improve quality of life, reducing secondary effects like fatigue and modifying the risk of depression [7,8]. Also, it has been demonstrated that a structured exercise program helps improve the functional capacity of cancer patients receiving chemo-radiotherapy [9]. The American College of Sports Medicine calls to avoid inactivity in all forms of cancer, even in those with difficult treatment or prognosis [10]. Exercise has been considered part of the complementary treatment for hematological patients, including psychology, palliative care, rehabilitation, thanatology, nutrition, etc. [11]. 

Hilfiker et al. in their meta-analysis, included 245 clinical trials comparing various procedures, which included exercise by patients during and after different treatments. They identified exercise as safe when it does not involve a lot of movement or physical exhaustion (aerobic, endurance, or combined). Even relaxation exercises, massages, and yoga act as protective factors at any stage of cancer treatment [12]. Similarly to solid tumors, the benefit of exercise has been proven in hematological diseases, especially in young individuals [13,14,15]. In line with this, most of the evidence was found in the pediatric population using resistance exercises, which allowed them to be included in the standard treatment. However, even when its impact is known, other benefits are still unknown when other variables are present, like anthropometry, body composition, quality of life, and physical performance. Another potential of exercise is to reduce the risk of sarcopenia or muscular problems, but the evidence is still limited, especially in the early stages of the treatment [16].

In general, studies that have evaluated the use of resistance exercises in hematologic patients have shown them to prevent the muscular deterioration associated with the disease. However, it is believed that cross-training, in addition to preventing muscular damage, could improve performance and other physical capabilities like core and general strength, stability, and joint mobility.

Therefore, a randomized pilot study was considered to research the effect of cross and resistance training on anthropometry, body composition, quality of life, and physical performance on ALL adult patients during the induction stage. The hypothesis suggested that implementing a cross-training routine would be feasible and safe for adult patients. All have an effect similar to or greater than resistance exercises compared to regular treatment.

## 2. Materials and Methods

Thirty-three patients with ALL “de novo” diagnoses were included; they had started the induction stage under the CALGB 10403 treatment scheme between May 2021 and September 2022 at the Department of Hematology of Hospital General de México “Dr. Eduardo Liceaga.”. The exclusion criteria were patients (1) with neutropenia, infections, and bleeding at admission; (2) who were nonmotile or unable to carry out exercise; (3) with a central nervous system disease that prevented movement; (4) with alterations of heart function; (5) with bone marrow or central nervous system relapse; or (6) with a referral from another hospital and who were treated at our service. Every patient provided their written informed consent for the implementation of the study and data collection. This prospective pilot, blinded, three-armed, open-label randomized study (1:1:1 ratio) was carried out following the Helsinki Declaration and approved by the Biosecurity, Ethics, and Research Committee of Hospital General de México “Dr. Eduardo Liceaga”, under the protocol number HGMDI/21/204/03/46. It was registered at ClinicalTrial.gov under the registration code NCT05059847.

### 2.1. Principal Outcomes

At admission, anthropometrics, body composition, quality of life, and physical performance data were recorded at the service (baseline). Every measurement was taken at four points: (1) baseline (at the beginning before the induction chemotherapy); (2) at hospital discharge after the first treatment scheme (+28 days); (3) upon completion of the first intensification scheme (+60 days); and (4) upon finishing the first consolidation scheme (+90 days). Overall, every patient had a three-month follow-up.

Physical performance was measured using five resistance and aerobic capacity tests that objectively evaluated the exercise. The resistance tests used were: (1) the “Sit-up test”, in which the patient was asked to lie down on a flat surface in a supine position and place their knees bent at 90 degrees with their hands clasped behind the nape while an assistant held their ankles, and they then had to raise the trunk to their knees as many times as possible within 30 s [17]; (2) the “Sit to stand test”, which consisted of sitting in and getting up from a chair (the seat was 44 cm in height) with the hands against the body as many times as possible within 30 s [18]; the (3) grip strength test—using a hand dynamometer Lafayette J00105 (Lafayette Instrument, Lafayette, IN, USA) the patient was asked to sit on a flat surface with their dominant arm at an angle of 90 degrees and then to clench their hand with as much strength as possible for 5 s. Three measurements were taken, and the average of these measurements and the average movement evaluation were used [19]. The following additional tests were carried out: (4) the “Get up and go” test, commonly used to evaluate agility and dynamic balance, in which the patient was asked to sit on a chair and then stand up and walk as fast as they could 3 m and back, and the time was measured from the moment they stood up from the chair until they sat back down [20]; and (5) the 6-min walk test, in which a hallway was prepared with markings on the floor every 2 m up to the 20-m mark, and the patient was asked to walk for 6 min as far as possible [21]. The oxygen saturation, blood pressure, and heart rate were measured before and after the test. The number of laps (meters) covered in those 6 min was estimated, as well as the rating of perceived exertion (RPE) caused by that activity (Borg scale).

The quality of life (%) was analyzed through the FACT-Leu questionnaire [22], including aspects of health related to quality of life being classified into five subscales: physical well-being (PWB), social (SWB), emotional (EWB), functional (FWB), and other factors.

Body weight and height were measured using a BAME brand (Mexico) mechanical scale with a calibrated stadiometer with a weighing capacity of up to 160 kg and a height scale of up to 1.95 m. The measurements for weight and height were made according to the manufacturer’s specifications. The values of BMI were calculated using the standard formula. The circumferences of the arm, waist, abdomen, hip, and calf were measured and rounded to the closest 0.1 cm using a Lufkin metal tape. This test was carried out at the beginning, during the two following sessions, and upon the completion of the intervention.

For the measurement of the body composition, an 8-electrode bioimpedance analyzer SECA mBCA 525 (SECA, Hamburg, Germany) was used, with the patient lying down in a supine position with arms and legs slightly separated from the body. Two electrodes were placed on each hand: one at the radiocarpal joint and the other at the carpometacarpal joint. On the feet, the electrodes were placed at the tibial-tarsal joint and at the tarsal bones. These electrodes were connected to the belt that would be placed around the legs of the patient.

The procedure took place with the patient having fasted; being adequately hydrated; without showing perspiration, fever, or feeling cold; without shoes, socks, or jewelry (bracelets, wristwatches, necklaces, or chains) on the arms, ankles, or other parts of the body; and without having carried out strenuous exercise 12 h before the study.

### 2.2. Exercise Intervention

The patients were assigned at random to any of the following groups: cross-training exercise (CEG), resistance exercise (REG), or control (CG). The exercise interventions took place during the hospital stay (at the induction chemotherapy) and on an outpatient basis (during the start of the intensification and consolidation stages) through 3 to 5 exercise sessions per week with a duration of 30 to 50 min each. The training sessions were supervised and documented by a specialized trainer. The resistance exercise protocols were based on previous studies and exercise guides for patients with hematological cancer [13,14,23,24,25].

The patients in the CEG carried out customized and supervised exercises that focused on improving joint mobility, joint stability, and the global and abdominal strength of the body. The patients would start with a 5-min warm-up that included stationary and dynamic exercises to elevate the body temperature. Subsequently, the main stage consisted of a 30- to 40-min workout at an intensity established by an RPE of 3–6 (equivalent to 50–75% of heart rate reserve) [26] that included seven different activities that used a broomstick (for example, mobility exercises for the shoulder joint or farmer walk) and their body weight (for example, air squats, step-ups, and half push-ups). The density of the sessions was organized according to the progress and condition of each patient, who carried out from 3 to 5 sets of 8 to 15 repetitions. The exercise progression was adjusted monthly until the three follow-up months were completed. During the two months as outpatients, the exercise routines were supervised through video call on an established schedule; when the patients were unavailable at the specified time, evidence of them carrying out the practices at home was requested. Since most of the monitoring occurred during the pandemic, a remote method was employed to avoid overcrowding and reduce the risk of infection.

The patients in the REG carried out customized and supervised resistance exercises for large muscle groups, including activities with body weight (for example, squats, lunges, and lumbar bridges) and exercises with lightweight accessories (for example, dumbbells). The prescribed exercise protocol included ten different exercises; the intensity, sets, and repetitions were adapted to the values retrieved from the RPE. The weight and difficulty were adjusted monthly. The patients received the necessary equipment to carry out the exercises at home (dumbbells and mats) from the person responsible for the protocol.

The CG included a low-intensity intervention consisting mainly of daily mobilization for at least 30 min to avoid depression and prostration.

Every patient received standard clinical care, including nutritional and psychological counseling by dietitians and psychologists specialized in hematologic patients. On a routine basis, to evaluate the safety of the patients, hematic biometrics analysis and vital signs were taken by the nursing team of the service. Patients omitted the exercise sessions when they found themselves in the following situations: platelets < 20 × 10^3^/µL, hemoglobin < 6.0 g/dL, temperature >38 °C, bleeding, or adverse effects post-chemotherapy.

### 2.3. Sample Size and Randomization

The sample size was calculated using a formula that found the difference between the measures of two independent groups. It was calculated using the software G. Power 3.1.9.2 using the standard means and deviations of the distance covered (in meters) of the “6-min walk” test carried out by the resistance exercise group and the control group post-intervention from the Alibhai et al. study [13]. Considering a statistical power of 0.8, an effect size of 0.69, and 20% of potential losses, a total of 38 patients per group was calculated for 114 patients. However, since this was a pilot study, it was assumed that 9% of the patients per group would be an adequate number of patients to be included [27]. Due to the situation during the pandemic and because the hospital would be considered a reference center for patients with COVID-19, our recruitment process stopped during the peaks of the 3rd, 4th, and 5th COVID-19 waves (around five months in total). Patients were randomly selected through a randomized list generated by computer software. The study was blinded to the attending physicians of each patient; only the patients and the staff in charge of the training sessions knew the group to which they belonged.

### 2.4. Statistical Analysis

The demographic characteristics were described as means (±SD) for quantitative variables and cases (*n*, %) for categorical variables. Every variable was analyzed via parametric analysis since they presented a normal distribution according to the Shapiro–Wilk test. The difference between the groups was calculated using analysis of variance tests (ANOVA for three groups) for the quantitative variables and chi-squared tests for the categoric variables. An ANOVA of repeated measures was used to evaluate the changes in the variables of body composition, physical performance, and quality of life during the three months of the study. For comparing variables pre- versus post-intervention, a Student’s *t*-test was used for related samples, and Cohen’s d formula represented the effect size. In addition, Pearson correlation and relative risk tests were carried out in the exercise interventions and the patient’s clinical prognosis. Lastly, the log-rank test determined the number of relapse cases within the groups for 500 days, including the study’s follow-up. All data were represented as means (±SD) and proportions (%). A value of *p* < 0.05 was considered to have statistical relevance. The statistical analysis was carried out using the statistical software SPSS version 25 (IBM Corporation, Armonk, NY, USA), and the figures were generated using GraphPad Prisma version 7 (GraphPad Software, Boston, MA, USA).

## 3. Results

A total of 50 patients were considered for the study. However, a total of 33 patients were selected and assigned to the groups CEG (*n* = 11), REG (*n* = 11), or CG (*n* = 11), which corresponded to 30.7% of the calculated sample size. After the three-month follow-up, only 54.5% (*n* = 18) completed the study. Nevertheless, only 6% (*n* = 2) could not meet at least two measurements; for this reason, the data of the remaining 18 patients were analyzed (Figure 1). The main reasons for not fulfilling at least two measures were: a decrease (*n* = 4), treatment desertion (*n* = 5), and loss of treatment follow-up due to the COVID-19 pandemic during the third wave (Omicron variant) that struck our country and the hospital. The demographic characteristics of the groups were similar among the groups (Table 1).

The pre-and post-intervention results of the changes in body composition, physical performance and strength, and quality of life are represented in Table 2, Table 3 and Table 4, respectively. The number of patients included in the analysis was also identified since some patients could not complete the three-month follow-up due to a decrease before consolidation.

Within the variables of body composition, when comparing the final variables among the groups, significant differences were noted. Nevertheless, within the groups, the CG was the group that presented more changes before versus after the follow-up of the study. In the CG, the variable of PFFM significantly decreased at the end of the survey (*p* = 0.02, Cohen’s d: −0.51), while the BFM (*p* = 0.015, Cohen’s d: 0.37), PBFM (*p* = 0.02, Cohen’s d: 0.51), and VF (*p* = 0.031, Cohen’s d: 0.48) appeared to have increased. For the REG, relevant differences were observed; on the one hand, the FFM decreased at the end of the follow-up (*p* = 0.048, Cohen’s d: −2.45), which was read as a clinically relevant reduction, while the VF increased in comparison to the base values (*p* = 0.024, Cohen’s d: 0.46). No significant difference was found in the CEG.

Significant differences were identified in the physical performance variables within and among the groups. At the end of the follow-up, the variables of the sit-up test (*p* = 0.009), the sit to stand test (*p* < 0.000), and the 6-min walk test (*p* = 0.004) presented significant differences when comparing both intervention groups with the CG. In contrast, in the “Get up and go” test, a significant difference (*p* < 0.000) was present only between the REG and the CEG. Within the groups, the CG only presented a statistically significant difference in the “Get up and go” test post-intervention (*p* = 0.05, Cohen’s d: −2.89). In the REG, the sit-up test (*p* = 0.02, Cohen’s d: 9.32), the sit to stand test (*p* = 0.009, Cohen’s d: 8.54), the “Get up and go” test (*p* = 0.029, Cohen’s d: −4.27), and the 6-min walk test (*p* = 0.007, Cohen’s d: 42.29) presented statistically significant differences in the post-intervention physical tests. For the CEG, only the variables of the it-up test (*p* = 0.009, Cohen’s d: 5.54), the sit to stand test (*p* = 0.003, Cohen’s d: 11.41), and the 6-min walk test (*p* = 0.006, Cohen’s d: 33.63) presented differences in the post-intervention physical tests that were both statistically and clinically significant.

No significant differences were found within the groups in the analysis of the quality of life variables. For the group study, only the CG presented a significant difference compared to the base values in the emotional well-being section (*p* = 0.02). The total values were above the base values for the CG and the REG. The CEG presented a total average value below the base values; however, the differences were statistically significant in either case.

A two-way repeated measures ANOVA test was performed on the main variables that were found to be significant and for which their change was associated with the type of intervention to which they were assigned. The variables that were analyzed according to the intervention were body composition: PFFM (F: 0.167, *p* = 0.848, np^2^ = 0.022), PBFM (F: 0.135, *p* = 0.875, np^2^ = 0.018), PSMM (F: 0.301, *p* = 0.744, np^2^ = 0.039), and visceral fat (F: 0.212, *p* = 0.811, np^2^ = 0.027); and physical performance: the sit-up test (F: 1.453, *p* = 0.265, np^2^ = 0.162), sit to stand test (F: 7.835, *p* = 0.005, np^2^ = 0.511), “Get up and go” test (F = 3.811, *p* = 0.046, np^2^ = 0.337), and 6-min walk test (F:6.432, *p* = 0.010, np^2^ = 0.479). It was observed that only the physical performance variables had statistical significance as well as an effect size considered to be mild to moderate when comparing the groups during follow-up. Figure 2 and Figure 3 graphically show the behavior of the variables throughout the study.

Based on the exercise interventions included in this study, Pearson’s r and relative risk (RR) values were estimated for the risk of relapse, failure at induction, and 500-day relapse (Table 5). During the study follow-up, only one patient reported an early relapse (<90 days), so it was decided to extend it up to 500 days since they continued to attend their follow-up appointments at the hospital and continued with their treatment. Based on the results, it was determined that none of the interventions had a statistical effect on clinical prognoses, and the correlations were very low.

Lastly, the relapse risk was analyzed using the log-rank test by comparing the patients who received exercise intervention during their induction scheme (68.57%; *n* = 24) (training routine) against the control patients (31.43%; *n* = 11 (no training routine)) (Figure 4a). However, it did not present significant relevance (Log-rank: 0.158). Furthermore, we analyzed the relapse risk according to the intervention the patients were assigned (Figure 4b). At the end of the 500-day follow-up, in the CG, 45.5% of the patients (*n* = 5) had a relapse compared to 25% (*n* = 3) of the REG group and 16.7% (*n* = 2) of the CEG group. Even though this was clinically relevant, no statistically significant differences were found (log-rank: 0.327).

## 4. Discussion

The main objective of this study was to analyze the independent effects of a cross-training exercise and resistance exercise routine on the physical performance, body composition, and quality of life of adult patients with ALL who underwent a standard induction scheme and compare them to the control group.

When analyzing the results, it was confirmed that both exercise interventions were safe and easy to implement. They presented no significant adverse events, allowing the patient to improve their physical performance and preserve their body composition.

The results could have been more consistent when evaluating the impact on quality of life with the different strategies. However, despite not being significant, a trend toward improvement was identified in the resistance exercise group, a situation that will be further evaluated in future essays.

When analyzing the exercise strategy, most changes were present in the body composition, for which the fat-free mass percentage decreased significantly compared to the base values (*p* = 0.02). At the same time, the BFM, PBFM, and visceral fat increased significantly (*p* = 0.015, 0.02, and 0.03, respectively).

Furthermore, in the intervention groups, the main significant changes in the resistance exercise group were observed in the reduction in the FFM (*p* = 0.048) and the increase in visceral fat (*p* = 0.024); in the CEG, body composition presented no significant changes in the initial values.

These data aligned with those described by Hartman et al., who evaluated an exercise program that focused on maintaining hand and leg function, stretching exercises, and short high-intensity exercises in children and adolescents undergoing induction therapy. Just like in our study, the BMI increased in the intervention group (∆1BMI = 1.53 DE) and the control group (∆1BMI = 1.38 DE). Nevertheless, this increase was not significant within the groups, and no differences were found when comparing the groups.

The percentage of body fat mass increased in both groups. Nevertheless, no differences were found among the groups during the treatment (∆1PBFM = 1.04 DE vs. ∆1PBFM = 1.56 DE, *p* = 0.25). While the percentage of fat-free mass was reduced in both groups, it presented no statistical significance (∆1PFFM = −0.61 DE vs. the control group ∆1PFFM = −0.12 DE, *p* = 0.16) [28].

These modifications to the adipose tissue can be explained by drugs used during the induction, such as steroids (dexamethasone and prednisone). These drugs are included in every induction scheme in high doses and have been linked to the development of obesity, especially in the pediatric population; this effect is explained by the increase in fatty acids in the circulation as well as the lipogenesis in the hepatocytes mediated by the synthetase fatty acid, which causes central obesity. In addition to this, the glucose metabolism is also affected due to the substitution of glucose production with glycogen [29,30,31]. Other changes related to the chronic use of steroids include behavior modifications, which increase the risk of depression and sedentarism, thereby increasing the risk of developing obesity for leukemia survivors [17].

Four out of five tests showed improvement in the intervention group regarding physical performance. This suggests that this strategy could improve the physical performance of the patients, even those with a medical condition presenting pain, chronic fatigue, and sarcopenia.

In leukemias, Byrant et al. evaluated the effect of exercise on the physical function of 17 adults with acute leukemia during induction therapy; their analysis focused on the performance of the “Get up and go”, 6-min walk, and pressure force tests without finding significant differences between the intervention group (exercise) and the control group (the intervention took place twice a day for four weeks) [13]. In contrast, Alibhai et al. found significant differences in the 6-min walk test when using a program of mixed exercises four to five times a week. When analyzing most studies that dealt with training by individuals with leukemia, the results remained inconclusive because they included few participants, and the exercise interventions were neither specific nor heterogeneous [17].

Another point of interest in exercise is the impact of this activity on the quality of life scales. Unlike what was previously supposed, there were no significant differences between the three intervention groups. We consider that the results were influenced by the social situation of the individuals, by their family network (since they are vulnerable individuals), and by the effect of the various medical and psychological interventions. Interestingly, when evaluating each questionnaire item, a significant difference in the control group was identified in the emotional well-being section. An important point regarding these scales is their effect on the health of the various quality of -life items; hence, when discussing leukemia, favorable changes in the disease are necessary to modify the quality of life.

Finally, we consider that this study, despite its limitations, highlights the importance of exercise in severe illnesses like acute leukemia. Although alterations in blood counts were shown, exercise routines could be implemented to improve the functional status and reduce the risk of adiposity following treatment. When considering the types of exercises, one of the strengths of this study was the intervention based on cross-training since it was the only intervention that did not require additional tools or weight to carry it out, and it could be conducted even remotely. Furthermore, the greatest weakness of this study was related to the recruitment during the pandemic of SARS-CoV2, which reduced patient flow and caused a halt during the waves with a more significant impact.

## 5. Conclusions

In conclusion, despite evidence pointing to certain types of tumors, exercise can impact the outcome, and it is necessary to gather further evidence for hematological malignancies. Based on our results, we recommend integrating supervised exercise routines, starting with induction therapy; such routines should include resistance or strength-based exercises since these are the critical components to maintaining physical and functional capabilities.

## Figures and Tables

**Figure 1 jcm-12-05008-f001:**
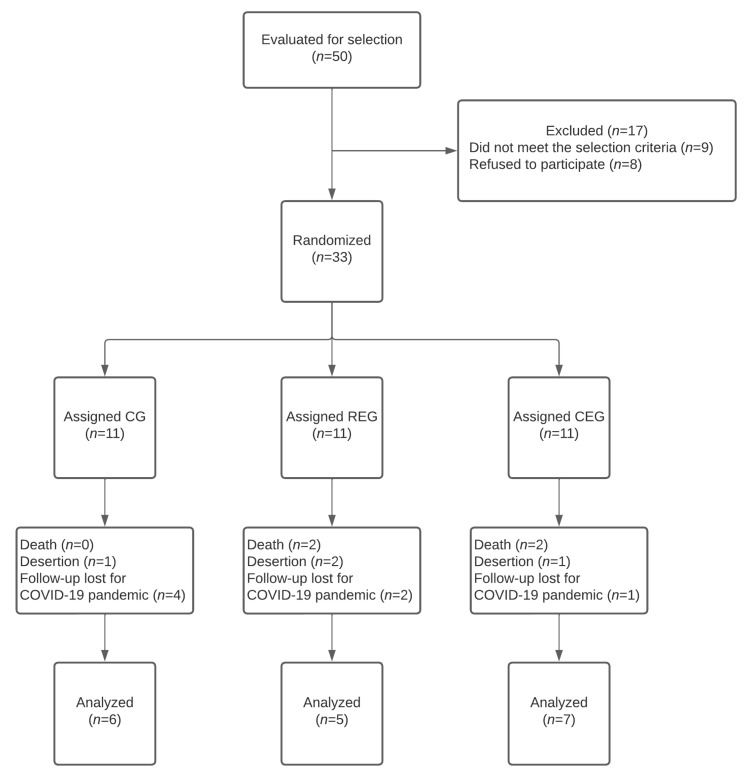
Flowchart of the patients’ recruitment and analysis. *n*: number.

**Figure 2 jcm-12-05008-f002:**
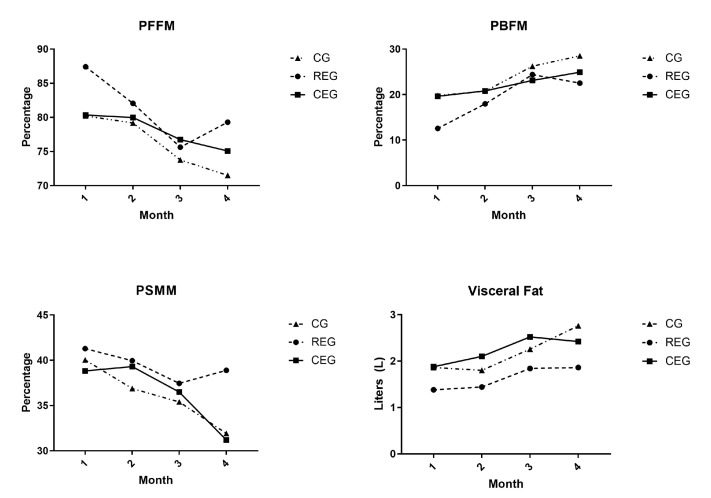
Repeated measures ANOVA for body composition variables.

**Figure 3 jcm-12-05008-f003:**
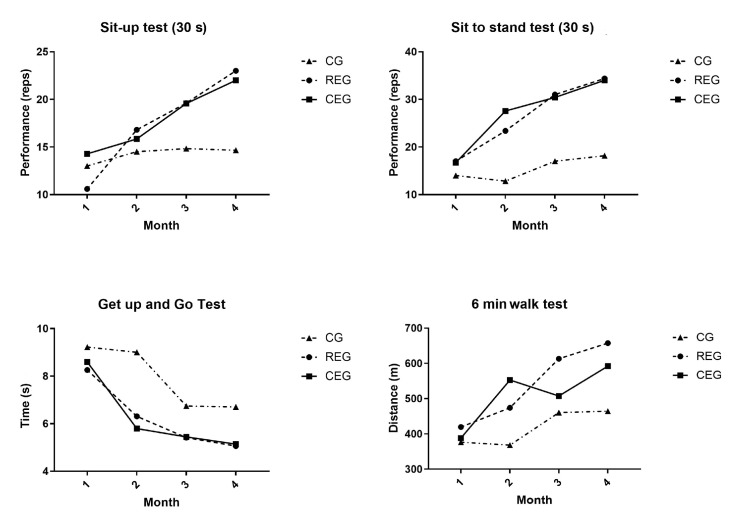
Repeated measures ANOVA for physical performance variables.

**Figure 4 jcm-12-05008-f004:**
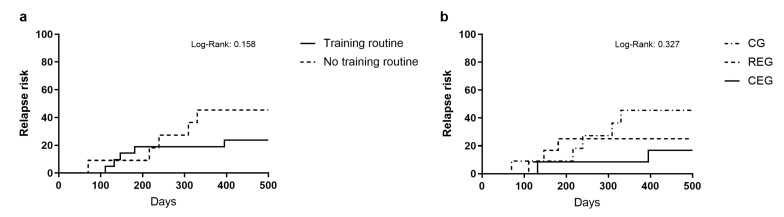
Kaplan–Meier diagram of patients with relapse classified by (**a**) Training routine and no training routine received and (**b**) type of exercise assigned.

**Table 1 jcm-12-05008-t001:** General characteristics of the study population.

	CG (*n* = 11)	REG (*n* = 11)	CEG (*n* = 11)	*p*-Value
**Age (years)**	28.0 (18–45)	22.5 (18–36)	20.5 (18–36)	0.166
**Sex (M:F)**	5:6	4:7	2:9	0.327
**Weight (kg)**	72.5 ± 18.53	75.29 ± 17.70	76.40 ± 26.22	0.904
**Height (m)**	1.65 ± 0.11	1.67 ± 0.09	1.67 ± 0.10	0.913
**BMI (kg/m^2^)**	26.28 ± 5.58	26.76 ± 6.03	26.91 ± 7.78	0.972
**MAC (cm)**	31.98 ± 4.96	30.65 ± 4.57	31.15 ± 6.02	0.831
**Waist circum. (cm)**	85.37 ± 23.53	89.43 ± 13.41	92.88 ± 13.41	0.643
**Abd. circum. (cm)**	95.60 ± 13.71	94.07 ± 15.57	95.95 ± 19.91	0.958
**Hip circum. (cm)**	100.24 ± 11.03	99.26 ± 11.38	97.92 ± 14.87	0.906
**Calf circum. (cm)**	33.91 ± 2.54	35.35 ± 4.22	34.77 ± 5.70	0.738
**FFM (kg)**	54.45 ± 10.85	60.78 ± 13.32	60.03 ± 19.07	0.551
**PFFM (%)**	77.47 ± 16.24	81.99 ± 12.47	80.08 ± 15.21	0.763
**BFM (kg)**	17.67 ± 13.85	14.92 ± 11.64	16.36 ± 12.56	0.873
**PBFM (%)**	22.52 ± 16.24	18.00 ± 12.47	19.91 ± 15.21	0.763
**SMM (kg)**	26.26 ± 6.64	29.31 ± 7.13	29.07 ± 10.24	0.621
**PSMM (%)**	37.49 ± 9.79	39.16 ± 5.97	39.46 ± 7.64	0.817
**Phase angle (°)**	6.12 ± 0.97	6.43 ± 0.77	6.34 ± 0.75	0.542
**Visceral fat (L)**	1.88 ± 1.64	1.66 ± 1.52	2.59 ± 2.31	0.457
**Sit-up test (reps)**	13.90 ± 6.80	12.91 ± 3.98	14.04 ± 4.66	0.843
**Sit to stand (reps)**	14.36 ± 2.29	15.75 ± 7.61	16.16 ± 6.63	0.761
**Grip strength test (kgf)**	25.18 ± 9.37	26.27 ± 9.28	29.84 ± 6.57	0.392
**“Get up and go” test (sec)**	9.30 ± 2.64	8.03 ± 1.69	8.13 ± 3.67	0.495
**6 min walk test (m)**	365.09 ± 86.40	443.66 ± 113.16	394.41 ± 100.42	0.184

M: male; F: female; BMI: body mass index; MAC: mid-arm circumference; Circum.: circumference; Abd: abdominal; FFM: fat-free mass; PFFM: percentage of fat-free mass; BFM: body fat mass; PBMF: percentage of body fat mass; SMM: skeletal muscle mass; PSMM: percentage of skeletal muscle mass. We used an ANOVA test for quantitative variables and a chi-squared test for qualitative variables. A *p*-value < 0.05 was considered as a statistical significance parameter.

**Table 2 jcm-12-05008-t002:** Intra- and inter-group comparison of changes in body composition at baseline and at the end of the study.

	CG (*n* = 6)			REG (*n* = 5)			CEG (*n* = 7)			*p*-Value ^b^
	Baseline	Post	*p*-Value ^a^	Baseline	Post	*p*-Value ^a^	Baseline	Post	*p*-Value ^a^	
**Weight (kg)**	74.58 ± 22.04	75.66 ± 18.66	0.633	73.86 ± 14.95	72.16 ± 16.46	0.335	67.60 ± 22.63	69.50 ± 27.50	0.449	0.883
**BMI (kg/m^2^)**	26.33 ± 6.95	26.54 ± 5.31	0.788	26.94 ± 5.59	26.43 ± 6.29	0.374	24.82 ± 6.45	25.44 ± 7.93	0.466	0.949
**MAC (cm)**	33.31 ± 5.94	30.45 ± 4.46	0.197	31.80 ± 3.63	31.46 ± 5.55	0.722	28.94 ± 4.58	29.22 ± 5.11	0.545	0.750
**Waist circum. (cm)**	82.2 ± 31.99	93.00 ± 13.00	0.269	89.08 ± 10.94	86.36 ± 12.57	0.111	87.81 ± 16.75	89.70 ± 21.24	0.456	0.808
**Abd. circum. (cm)**	95.25 ± 17.44	97.50 ± 14.27	0.439	92.50 ± 11.91	90.12 ± 14.37	0.349	91.68 ± 18.04	93.07 ± 20.66	0.464	0.774
**Hip circum. (cm)**	100.35 ± 13.57	102.78 ± 12.02	0.079	98.88 ± 7.85	99.48 ± 9.06	0.699	94.12 ± 14.46	94.71 ± 13.69	0.822	0.495
**Calf circum. (cm)**	34.33 ± 2.42	36.50 ± 4.05	0.180	35.12 ± 3.05	36.34 ± 3.15	0.054	32.62 ± 4.42	33.35 ± 5.23	0.213	0.372
**FFM (kg)**	56.84 ± 10.53	52.70 ± 10.07	0.153	64.68 ± 14.32	55.99 ± 10.80	0.048	53.33 ± 16.92	50.35 ± 15.43	0.081	0.753
**PFFM (%)**	80.21 ± 18.72	71.50 ± 15.41	0.020	87.44 ± 7.46	79.30 ± 10.67	0.119	80.35 ± 17.67	75.07 ± 14.99	0.079	0.667
**BFM (kg)**	16.93 ± 16.71	22.86 ± 14.81	0.015	9.17 ± 5.29	16.16 ± 9.77	0.079	14.26 ± 13.02	19.18 ± 15.56	0.092	0.732
**PBFM (%)**	19.78 ± 18.72	28.50 ± 15.41	0.020	12.56 ± 7.46	22.50 ± 11.29	0.066	19.64 ± 17.67	24.92 ± 14.99	0.079	0.783
**SMM (kg)**	28.10 ± 6.62	23.73 ± 5.73	0.094	30.80 ± 7.48	27.68 ± 4.94	0.168	24.81 ± 9.19	23.55 ± 9.28	0.238	0.578
**PSMM (%)**	40.05 ± 11.56	31.92 ± 7.59	0.056	41.30 ± 4.11	38.90 ± 5.76	0.420	38.83 ± 9.02	31.19 ± 15.46	0.134	0.467
**Phase angle (°)**	5.87 ± 1.27	8.00 ± 14.67	0.792	6.66 ± 0.47	4.00 ± 2.64	0.201	6.43 ± 0.73	7.66 ± 24.34	0.494	0.194
**Visceral fat (L)**	1.86 ± 2.10	2.76 ± 1.64	0.031	1.38 ± 0.99	1.86 ± 1.14	0.024	1.88 ± 1.79	2.42 ± 2.59	0.275	0.754

BMI: body mass index; MAC: mid-arm circumference; Circum: circumference: FFM: fat-free mass; PFFM: percentage of fat-free mass; BFM: body fat mass; PBMF: percentage of body fat mass; SMM: skeletal muscle mass; PSMM: percentage of skeletal muscle mass. The ANOVA test for quantitative variables and the Student’s *t*-test for related variables were used to complete the analysis. ^a^ Intra-group *p*-value; ^b^ inter-group *p*-value. A *p*-value of <0.05 was considered a statistical significance parameter.

**Table 3 jcm-12-05008-t003:** Intra- and inter-group comparison of physical performance at baseline and at the end of the study.

	CG (*n* = 6)			REG (*n* = 5)			CEG (*n* = 7)			*p*-Value ^b^
	Baseline	Post	*p*-Value ^a^	Baseline	Post	*p*-Value ^a^	Baseline	Post	*p*-Value ^a^	
**Grip strength test (kgf)**	24.36 ± 4.16	27.42 ± 3.30	0.535	30.65 ± 4.67	33.33 ± 3.64	0.095	28.24 ± 2.77	28.83 ± 2.80	0.547	0.482
**Sit-up test (reps)**	13.00 ± 3.21	14.66 ± 1.72	0.410	10.60 ± 2.24	23.00 ± 1.30	0.020	14.28 ± 1.96	22.00 ± 1.91	0.009	0.009 ^†^
**Sit to stand test (reps)**	14.00 ± 1.00	18.16 ± 2.00	0.090	17.00 ± 5.12	34.4 ± 3.18	0.009	16.71 ± 3.04	34.00 ± 1.55	0.003	0.000 ^†^
**“Get up and go” (sec)**	9.22 ± 1.18	6.75 ± 0.28	0.050	8.26 ± 0.89	5.06 ± 0.23	0.029	8.59 ± 1.77	5.14 ± 0.20	0.084	0.000 ^‡^
**6-min walk test (m)**	387.00 ± 36.37	463.00 ± 30.73	0.166	419.00 ± 21.24	657.40 ± 42.30	0.007	388.28 ± 44.67	592.14 ± 28.80	0.006	0.004 ^†^

The ANOVA test for quantitative variables and the Student’s *t*-test for related variables were used to complete the analysis. ^a^ Intra-group *p*-value; ^b^ inter-group *p*-value. A *p*-value of < 0.05 was considered a statistical significance parameter. ^†^ Statistical difference (*p* < 0.05) between both exercise interventions and the control group. ^‡^ Statistical difference (<0.05) between resistance exercise and cross-training exercise.

**Table 4 jcm-12-05008-t004:** Intra- and inter-group comparison of quality of life through FACT-Leu questionnaire.

	CG (*n* = 6)			REG (*n* = 5)			CEG (*n* = 7)			*p*-Value ^b^
	Baseline	Post	*p*-Value ^a^	Baseline	Post	*p*-Value ^a^	Baseline	Post	*p*-Value ^a^	
**PWB (0–28)**	23.35 ± 6.41	24.15 ± 3.92	0.796	22.00 ± 2.00	21.80 ± 3.49	0.898	20.93 ± 3.08	19.00 ± 4.00	0.194	0.084
**SWB (0–28)**	22.77 ± 5.23	24.85 ± 2.56	0.231	17.01 ± 5.89	18.66 ± 9.06	0.639	21.21 ± 4.75	18.18 ± 7.39	0.391	0.195
**EWB (0–24)**	14.50 ± 7.89	20.16 ± 4.35	0.020	18.16 ± 2.79	21.00 ± 2.34	0.103	14.14 ± 4.52	15.14 ± 6.38	0.731	0.260
**FWB (0–28)**	22.08 ± 4.15	23.16 ± 4.75	0.560	18.30 ± 6.05	18.0 ± 2.23	0.909	20.37 ± 5.06	16.28 ± 4.60	0.097	0.382
**Others (0–64)**	49.71 ± 11.21	49.50 ± 11.29	0.975	51.92 ± 6.84	50.80 ± 13.00	0.888	46.45 ± 6.72	40.57 ± 12.17	0.386	0.151
**Total (0–176)**	133.59 ± 24.68	141.85 ± 16.61	0.424	127.40 ± 19.64	130.26 ± 19.24	0.835	124.31 ± 14.21	108.90 ± 26.47	0.207	0.340

PWB: physical well-being; SWB: social well-being; EWB: emotional well-being; FWB: functional well-being. The ANOVA test for quantitative variables and the Student’s *t*-test for related variables were used to complete the analysis. ^a^ Intra-group *p*-value; ^b^ inter-group *p*-value. A *p*-value of < 0.05 was considered a statistical significance parameter.

**Table 5 jcm-12-05008-t005:** Relative risk and correlation between intervention groups and principal clinical outcomes.

	MRD + 45			Induction Response			Relapse		
	r	RR (IC95%)	*p*-Value	r	RR (IC95%)	*p*-Value	r	RR (IC95%)	*p*-Value
**CG** **(*n* = 6)**	0.03	0.889 (0.245–3.226)	0.619	0.25	0.480 (0.189–1.220)	0.138	0.07	0.802 (2.95–2.180)	0.479
**REG** **(*n* = 5)**	−0.09	1.333 (0.422–1.679)	0.453	0.03	0.917 (0.349–2.406)	0.576	−0.06	1.200 (0.407–3.539)	0.530
**CEG** **(*n* = 7)**	0.06	0.800 (0.224–2.855)	0.547	−0.10	1.375 (0.460–4.108)	0.424	−0.19	2.000 (0.529–7.557)	0.236

MRD: minimal residual disease; RR: relative risk; CG: Control Group; REG: Resistance Exercise Group; CEG: Cross-training Exercise Group. Values are shown as RR (CI 95%). A *p*-value < 0.05 was considered a statistical significance parameter.

## Data Availability

Data is available on request due to restrictions of privacy and confidentiality manners. The data presented in this study is available on request from the corresponding author. The data are not publicly available due to confidentiality restrictions of the Biosecurity, Ethics, and Research Committee of Hospital General de México “Dr. Eduardo Liceaga.”.

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
