# Peer review of "Cross-Training and Resistance Training in Adults with Type B Acute Lymphoblastic Leukemia during the Induction Phase: A Randomized Blind Pilot Study"

_jcm, 2023, doi:10.3390/jcm12155008_

Round 1

Reviewer 1 Report

According to the authors, this was a pilot study to compare the effects of cross-training vs. endurance(?) training or a control group in adult patients with ALL who were in the induction phase. The manuscript has various (serious) deficiencies, which are described in more detail below and are the reason why an extensive revision of the manuscript is necessary.

There are many inconsistencies throughout the manuscript, with only a few examples given here as examples:

11. The title says "Cross-Training and Resistance Training" but according to the text a "cross-training or an endurance training (EEG) was performed. The exercises described in the EEG are rather strength exercises (exercises with dumbbells).

2 2.  The numbers in the flow chart of the patient's recruitment and analysis (page 6) do not match or are not plausible, e.g. n = 50 minus n = 15 equals 33? In the EEG n = 11, 7 patients drop out and 5 remain (= 4?), and the gender distribution according to Table 1 (page 6) is 4:8, resulting in a total of n = 12 (also applies to the CEG).

·       The introduction lacks a common thread. The transitions between the sentences are partly not comprehensible/logical. While depression is not mentioned in the side effects of ALL treatment (page 2, line 49 to 57), the following paragraph describes in detail what effects sport has on depression. Some of the references do not refer to cancer patients, although there are numerous studies showing positive effects of exercise on depression. In addition, depression as well as other symptoms are not included in the study. Consequently, this is irrelevant.

Although this study was conducted during the acute phase (under therapy), it is pointed out in the introduction that sport has a preventive effect and also shows positive effects in rehabilitation (page 2, line 62-64). Both are irrelevant for this study. There are many studies also in leukemias and other hematological diseases that have conducted investigations under therapy, which would be more appropriate at this point. 

·       The tests mentioned for measuring physical performance are confusing in part because they were named or performed differently than standardly used. This makes comparability with other study results difficult. Examples:

o   Test 1 is a classic "sit-up" but is not named here, only the execution is described. In Table 1 it is then called "Abdominal test".

o   Test 2 is called "Sit-ups test" but is the so-called "Sit to stand test", which is performed for 60 s or 30 s as standard. Why 45 s were used here is not explained.

o   Test 4 is titled "Get up and walk" in Table 1 it is then called "Get up and go" which would be more correct (or Timed up & go). However, this is a test to measure mobility and fall risk and is often used in geriatrics. Therefore, this is not a strength or endurance test as assigned (page 3, line 111)! Why this test was used in the cohort remains unclear?

o   Test 5: The distance is extremely short (10 m), which means that the patients have to turn around very frequently, which can lead to distortions (comparability with other studies is lost). According to the literature, the subjects have the task to walk as far as possible during the 6 min walk test, a constant speed is not mandatory according to the literature (also depends on the performance, pain may occur in the course which leads to a slowdown ...). Furthermore, the BORG scale does not measure fatigue but the "Rating of perceived exertion" (RPE).

o   References to the tests used are completely missing.

o   It is unclear what is meant by "Speed of displacement" Table 1 bottom row and where this variable comes from.

o   It is questionable what is meant by "squat test" on page 10, line 288?

·       Described are 4 measurement time points (page 3, line 105 to 109), although it is not entirely clear whether the baseline measurement = time point 1 (at the beginning, before the induction chemotherapy). In the analysis, only 2 time points (baseline and post) are compared.

·       Page 4, line 175 to 176 it is written that the training intensity in the CEG was 60-70% of their maximum cardiac capacity, based on the results of the initial tests. In the test description there is no test with which the maximum cardiac capacity was determined.

·       Page 5, line 187 to 188 it says that the intensity, were adapted to the values documented during the initial tests. In the test description there are no corresponding tests (determination of 1-repeat maximum).

·       In a pilot study with 18 patients in three groups, the calculation and discussion of relapse risk seems inadequate.

Minor Comments

·       It would be beneficial if consistent terms were used. Some examples: quality of life vs. life quality, strength vs. restistance, exercise vs. physical therapy.Figure 1: letzte Zeile “Randomized” hier eher unpassend  besser „analyzed“

·       Some sentences are very long and should be split (e.g. page 2, line 78 to 82, page 3, line 90 to 93).

·       The order of the data, which were collected and evaluated, should always be given in the same order. Introduction: anthropometry, body composition, life quality, and physical performance (page 2, line 84 to 85)

o   Methods: demographic data, life quality, body composition, and physical performance (page 3, line 105 to 106)

o   Following, starting with physical performance gestartet, then quality of live (wording!), then demographic data finally body composition.

o   The order of the tests as described on page 3, line, differs in the order compared to Table 1.

·       Table 1: Heading: "Demographic characteristics of the study population", Table 1 also contains information on body composition and physical performance (order again different from text). It is questionable, since almost half of the participants are lost, what significance these data have.

·       Page 3, line 143 to 145: cutoff points proposed by the World Health Organization. What is meant by this? In the text there are only mean values and standard deviations.

·       Page 4, line 149 to 158: This part can be shortened to a few sentences, since the measurements are usually made according to the manufacturer's specifications. It should be added which parameters are determined.

·       Abbreviations used in the text should be introduced the first time they are used

Reviewer 2 Report

Gallardo-Rodriguez A.G. et al in the article titled `Cross-Training and Residence Training in Adults with Type B Acute Lymphoblastic Leukemia during the Induction Phase. A randomized Blind Pilot Study` describe a positive effect of different cross-training activities in recovering patients.

The text is written well, the ideas well organized and the quality of the tables and figure are of a good quality.

The authors also compare the previous work done by other groups and show the results of their randomized study.

The limitation of the study is as the authors themself stress is the period in which the study was done, during the Omicron wave in Mexico. The limitation refers also to the number of the patients. The suggestion would be also maybe to select few groups of patients with similar disease stages and then to compare the achieved effects, since the authors reported also the loss of some patients during different periods.

The article brings an interesting and important approach into recovery regiment of ALL patients.

Author Response

Response to Reviewer 2 Comments

Point 1: The suggestion would be also maybe to select few groups of patients with similar disease stages and then to compare the achieved effects, since the authors reported also the loss of some patients during different periods.

Response 1: We agree with your comments. We stratified our population only to acute lymphoblastic leukemia adult patients in induction therapy as it has not been studied before. It would be interesting to also analyze the patients in other moments of the treatment and compare them with our population,